# Chinese Pedigree with Hereditary Gastrointestinal Stromal Tumors: A Case Report and Literature Review

**DOI:** 10.3390/ijms24010830

**Published:** 2023-01-03

**Authors:** Qichao Ge, Yang Liu, Fan Yang, Guangwei Sun, Jintao Guo, Siyu Sun

**Affiliations:** 1Department of Gastroenterology, Shengjing Hospital of China Medical University, 36 Sanhao Street, Shenyang 110004, China; 2Innovative Research Center for Integrated Cancer Omics, Shengjing Hospital of China Medical University, Shenyang 110004, China; 3Innovative Engineering Technology Research Center for Cell Therapy, Shengjing Hospital of China Medical University, Shenyang 110022, China

**Keywords:** gastrointestinal stromal tumor, whole-exome sequencing, germline mutation, familial disease, precision oncology

## Abstract

Familial gastrointestinal stromal tumor (GIST) is a rare autosomal dominant genetic disorder with only a few affected families reported to date. Here, we report a case of familial GISTs harboring a novel germline mutation within exon 18 of *KIT*. A 58-year-old male patient presented with gastric subepithelial lesions accompanied by cutaneous hyperpigmentation, which were subsequently diagnosed as multinodular GISTs. Endoscopic surgery was initially conducted to remove the larger lesions, and pathological examinations were then conducted for the diagnosis of GISTs. Family history revealed that some other family members had similar cutaneous pigmentations. Whole-exome sequencing was used to search for potential driver mutations, and Sanger sequencing was used for mutation validation. A novel primary driver mutation of *KIT* (c.G2485C, p.A829P) was detected in these hereditary GISTs, which has been reported in some targeted chemotherapy-resistant GISTs. Cell models were subsequently established for the rapid screening of candidate drugs and exploring potential mechanisms. This mutation could lead to cell proliferation and imatinib resistance by ligand-independent activation of KIT; however, ripretinib administration was identified as an applicable targeted therapy for this mutation. The mutation activated the JAK/STAT3 and MAPK/ERK pathways, which could be inhibited by ripretinib administration. To the best of our knowledge, this is the first report of the KIT-A829P mutation in familial GISTs, complementing the pathogenesis of familial GISTs and providing valuable information for the precision treatment of this disease.

## 1. Introduction

Gastrointestinal stromal tumors (GISTs) originating from Cajal cells are common mesenchymal tumors in the digestive system and are mainly located in the stomach and small bowel [1]. GISTs are mostly caused by *KIT* mutations (present in approximately 75–80% of cases) and *PDGFRA* mutations (in approximately 10% of cases). Molecular sub-classification of GISTs also incorporates succinate dehydrogenase complex (SDH) deficiency, *BRAF* mutation, and genetic alterations in the RAS-mitogen-activated protein kinase (MAPK) pathway [2]. The clinical manifestations of GIST mainly include abdominal distension, abdominal pain, and abdominal mass [3]. The gold standard for a clinical diagnosis of GIST is a pathological diagnosis, including tumor cytomorphologic examination, immunohistochemistry, and molecular diagnosis. The immunochemical outcomes of GIST generally demonstrate positivity for KIT (CD117) and DOG-1 [4]. Genetic sequencing can be considered for wild-type and some difficult cases, especially for secondary drug resistance or hereditary cases.

Familial GIST is extremely rare and is mainly driven by germline mutations of related genes. Similar to GIST caused by somatic mutations, *KIT* and *PDGFRA* are the most common mutated genes in familial GIST [5]. The clinical manifestations of hereditary GIST are generally early-onset and involve multiple tumors, with a few cases accompanied by skin melanosis and mastocytosis [6,7]. Multiple GISTs with skin pigmentation are typically associated with mutations in the c-KIT- or NF-1-encoding gene, and the two most common KIT mutations are V559A and K642E [6,8,9,10]. Clinical chemotherapies for GIST are mostly tyrosine kinase inhibitors (TKIs), and mutations in different regions or forms lead to differences in the sensitivity of GIST to TKIs, regardless of the somatic or germline mutations of the driver genes. A missense mutation at the kinase activation loop of *KIT* (c. 2485G > C) that is associated with hyperpigmentation and lentigines was reported, but this mutation has not been detected in familial GISTs to date [11,12,13].

Here, we report a Chinese pedigree with progressive multiple GISTs and cutaneous hyperpigmentation, in which the same missense mutation of *KIT* was confirmed. This case highlights that the determination of GIST molecular subtypes is crucial for the accurate evaluation and treatment of patients.

## 2. Results

### 2.1. Case Presentation

In May 2016, a 58-year-old male patient was referred to Shengjing Hospital of China Medical University due to multiple gastric subepithelial lesions. An endoscopic ultrasound (EUS) showed that all lesions, some of which were exophytic, originated from the muscularis propria. The three largest lesions were located on the greater curvature side of the antrum–gastric body junction (one lesion) and the lesser curvature side of the gastric antrum (two lesions). The section sizes were 23 mm × 20 mm (Figure 1a), 30 mm × 20 mm (Figure 1b), and 10 mm × 8 mm (Figure 1c), respectively. The remaining lesions were in the gastric fundus, body, and antrum, with diameters of approximately 6–9 mm. Additionally, cutaneous hyperpigmentation and nevi 1–3 mm in diameter were noted, particularly around the mouth and on the hands and abdomen (Figure 1f; P1), but not on the lips or the oral cavity. Subsequently, endoscopic full-thickness resection (EFTR) and endoscopic submucosal dissection (ESD) were performed for the three larger lesions, and an over-the-scope clip system and clips were used to close the gastric wall defects. Histopathological examination (Figure 1e) revealed spindle cells arranged in the fascicles, and the mitotic images were not prominent. Immunohistochemistry showed positivity for KIT/CD117, DOG-1, and CD34 (Figure 1e) and negativity for calponin, smooth muscle actin (SMA), and S-100 (Appendix A). The Ki67 proliferation index was 5%. Pathological examination confirmed a diagnosis of multiple GISTs. For the remaining unresected stromal tumors, the proband took imatinib (400 mg/day) regularly for one month after endoscopic surgery, but no obvious therapeutic effect was observed. Notably, although we administered 400 mg of imatinib daily to the proband for one month, the patient complained of nausea, and the computed tomography scan indicated more enlarged nodules and increasing nodules. The proband refused to receive continuous chemotherapy but received close follow-up.

According to the patient’s statement, two other family members also suffered from cutaneous hyperpigmentation. Based on these clinical presentations, we conjectured that this may be a rare familial genetic disorder. Therefore, the family was followed up in an intensive study. The proband’s brother (P4) was also found to have multiple gastric subepithelial lesions, at the age of 54 years in 2016. The proband’s niece (Figure 1d,f; P2) was subsequently diagnosed with gastric GIST with cutaneous hyperpigmentation, at the age of 35 years in 2020, and was treated by EFTR in 2022. The daughter of the proband’s niece presented with hyperpigmentation on the toes, at the age of 11 years in 2022 (Figure 1f; P3), but gastrointestinal lesions were not detected.

During the follow-up period, the proband was regularly evaluated by CT. Comparing the CT images from 2019, 2021, and 2022 (Figure 2a), the proband was found to have enlarged gastric subepithelial lesions (the largest tumor progressed from 2.5 cm to 3.8 cm in diameter). Other multiple exophytic growing gastric nodules were also found during enlargement by endoscopy. Considering the progression of diffuse gastric lesions, the proband underwent a total gastrectomy in 2022. The gross specimen of the proband showed multiple diffuse extraverted stromal tumors in the entire stomach (Appendix A). The proband’s niece also showed progression of the gastric lesions, so ESD was performed. A schematic of the diagnosis and treatment strategy timeline for this family is presented in Figure 2b.

### 2.2. DNA Analyses

Whole-exome sequencing was conducted to detect a genetic defect in the family using peripheral blood DNA and tumor tissues. The results showed a point mutation in exon 18 of *KIT* (GCT > CCT, Ala > Pro) at codon 2485, which has not been reported in familial GISTs to date. This novel mutation was also detected, using targeted Sanger sequencing, in the formalin-fixed paraffin-embedded samples and EUS-fine needle aspiration tissues of the proband and normal tissues (oral epithelial cells) from all members of the family with cutaneous hyperpigmentation (Figure 1g,h) [14,15]. The mutation was highly coincident with the phenotypes, suggesting a germline origin (Figure 3).

### 2.3. Candidate Targeted Drugs Screening

To find a viable treatment, we established a cell model to conduct functional verification of this mutation and assess its sensitivities to the currently widely used KIT-targeted drugs. Briefly, KIT-expressing plasmids with pcDNA3.1(+), including the wild type and three mutational types (V559A, K642E, and A829P) were constructed. The plasmids were then transfected into the HEK 293T cell line for functional validation. Immunocytochemistry showed that the KIT protein was successfully expressed in the HEK 293T cells (Figure 4a). Sanger sequencing and Western blotting were used to verify the overexpression efficiency of these KIT mutations (Figure 4b,c). Ligand-independent experiments identified that *KIT*-A829P could lead to an auto-activation of the KIT protein in the absence of stem cell factor (SCF, the ligand of KIT) (Figure 5a). Cell viability experiments showed that *KIT*-A829P promoted cell proliferation (Figure 5b) and led to imatinib resistance (250 nM) (Figure 5b). Drug-sensitivity tests also showed that avapritinib with a dosage of 40 nM could not significantly inhibit cell proliferation because of its weak ability to inhibit KIT auto-activation (Figure 5b,c). Ripretinib, with a dosage of 150 nM, was ranked as the optimal drug for significantly inhibiting the cell proliferation and sustained phosphorylation of these three germline mutations (Figure 5b,c). According to the immunoblotting results, the JAK/STAT3 and MAPK/ERK signaling pathways were significantly activated by the *KIT*-A829P mutation.

## 3. Discussion

KIT is a transmembrane complex containing five domains that are encoded by a total of 21 exons [16,17]. The extracellular receptor region of the receptor tyrosine kinase (RTK) encoded by the KIT gene consists of five immunoglobulin-like domains (D1–D5) responsible for the recognition of specific stem cell factor ligands. When the receptor region binds to the ligand, a dimerization cascade reaction will occur, resulting in a conformational change and the phosphorylation of the entire RTK, which activates the downstream signaling pathway to produce functions. The juxtamembrane domain of the RTK encoded by exon 11 normally inhibits protein activation. Only when the RTK binds to the ligand and dimerizes does this inhibition release with the conformation of the protein, resulting in phosphorylation [17]. However, mutations in this domain will weaken or eliminate the inhibition, leading to a ligand-independent activation of the RTK. Mutations in the *KIT* oncogene have been reported in several tumors, including multiple GISTs [6,18]. The most common somatic mutations of *KIT* are in-frame deletions involving codons 557 and 558 of exon 11. Other primary somatic *KIT* genetic aberrations refer to exons 9, 13, and 17. The activation-loop domain is encoded by exon 17, where mutations can also lead to ligand-independent activation of this protein. In addition, several other germline mutations associated with GIST have been reported in recent years, although familial GIST remains rare among gastrointestinal tumors [10]. Multiple GISTs with cutaneous pigmentations are commonly associated with *KIT* or *NF-1* gene mutations [8,9,19,20]. The most common two mutations of KIT in familial GISTs are V559A and K642E. However, to the best of our knowledge, this is the first report of GISTs caused by germline mutations in *KIT* exon 18. Apart from mutations in mastocytosis and GISTs, mutations of *KIT* in melanomas have been found throughout the coding regions, with increased frequency in the juxta-membrane domain and tyrosine kinase domains. Germline *KIT* mutations are reported to be associated with other pigmented disorders, often accompanied by GISTs or mastocytosis [11,12]. A recent case report revealed a Chinese pedigree associated with progressive hyperpigmentation and generalized lentigines, carrying the same germline mutation of KIT (A829P), which suggested its role in the regulation of the proliferation and melanin production of the melanocytes [13].

According to the characteristics of this family, stromal tumors caused by KIT-A829P have the features of early onset and rapid progression. Moreover, this mutation could lead to whole stomach involvement in the later stage of the disease. Therefore, the rapid identification of sensitive therapeutic targets is the key to the subsequent accurate management of this family. Owing to its rarity, there is limited research on the TKI sensitivities of A829P variant-associated GISTs. The molecular mechanisms of *c-KIT* mutant GISTs typically involve activation of the PI3K-AKT-mTOR, RAS-RAF-ERK, and JAK/STAT signaling pathways, which influence the proliferation, survival, and anti-apoptosis activity of tumor cells [3]. The missense mutation identified in this study would lead to an amino acid substitution from alanine to proline at position 829 (A829P) on the kinase activation loop of KIT, generating a unique sequence “Pro-Arg-Leu-Pro”, which was postulated to serve as an Src homology 3 (SH3) domain-binding motif and activate the MAPK signaling cascade. This mutation was also reported to be associated with the acquired drug resistance during long-term treatment with imatinib mesylate in GISTs [21,22]. According to the results of the cell model, this mutation could lead to the constitutive activation of the KIT and the significant activation of the JAK/STAT and MAPK/ERK signaling pathways, which might contribute to the generation and progression of the multiple GISTs in the present family. Moreover, the GISTs driven by KIT-A829P appeared to be resistant to imatinib because of the weak inhibition of KIT activation, whereas ripretinib might be a recommended targeted treatment for significant inhibition of KIT activation and downstream signaling pathways.

Drug development in GIST has been oriented toward exploiting the high reliance on *KIT/PDGFRA* oncogenic signaling as a therapeutic vulnerability. Importantly, the *KIT* and *PDGFRA* genotypes predict imatinib activity, thereby providing very valuable clinical information [3,23]. The present drug experiment identified that KIT-V559A was sensitive to imatinib. Genetic alterations involving *KIT* exon 11 predict deeper and prolonged responses. GISTs harboring mutations in *KIT* exon 13 and 17 have been predicted to show less sensitivity or resistance to imatinib. The TKIs sunitinib and regorafenib are the standard second and third lines of treatment, respectively [24,25]. In the context of intratumoral heterogeneity, sunitinib and regorafenib appear to only partially inhibit the KIT kinase activity of imatinib-resistant subclones within their respective inhibitory profiles [26]. Therefore, specific treatment strategies in GIST will need to overcome tumor heterogeneity according to precise molecular diagnosis.

Ripretinib is an orally available type II switch-control TKI designed to inhibit the full spectrum of *KIT* and *PDGFRA* mutations and, therefore, has emerged as an innovative therapeutic approach against the heterogeneity of mechanisms of resistance [27,28]. Clinical trials conducted in advanced GIST patients illustrated that ripretinib significantly improved median progression-free survival compared with placebo (6.3 months vs. 1 month, respectively) [29,30,31]. In vitro, ripretinib exhibits potent anti-neoplastic effects following binding to KIT and PDGFRA receptors with mutations in exons 9, 11, 13, 14, 17, and 18 and exons 12, 14, and 18, respectively. Moreover, the ripretinib safety profile was favorable, and the side effects were mostly low-grade and manageable. Avapritinib is a highly potent and selective KIT and PDGFRA activation loop mutant kinase inhibitor, which is commonly used for the treatment of unresectable or metastatic GISTs carrying *PDGFRA* exon 18 mutations. Avapritinib also potently inhibits KIT N822K mutant autophosphorylation (half-maximal inhibitory concentration (IC50) = 40 nM), downstream signaling, and cellular proliferation (IC50 = 75 nM) [32]. However, avapritinib could not inhibit the auto-activation of KIT-A829P with a concentration of 40 nM in the present study. These findings indicated that the A829P mutation of KIT might disturb the interaction interface of specific drugs with the kinase; thus, ripretinib could be the optimal therapy for this type of pedigree in the future.

There are some limitations of this study. The mechanistic experiments are not sufficiently strong to prove that KIT-A829P drives tumorigenesis and influences the prognosis of this hereditary disease. Further exploration, such as constructing a transgenic mouse model, is ongoing to gain an in-depth understanding of the genotype–phenotype interaction.

## 4. Conclusions

Our study characterized a rare mutation of KIT (A829P) in familial GISTs. This novel mutation led to cell progression and TKI resistance due to the constitutive activation of KIT. Drug-sensitivity experiments identified ripretinib administration as potentially the most applicable targeted therapy for this family, demonstrating GISTs to be notable candidates for precision medicine.

## 5. Methods and Materials

### 5.1. Clinical Diagnosis and Treatment

Conventional endoscopy and endoscopic ultrasound were used in the imaging diagnosis of stromal tumors. Further pathological examinations were conducted according to the Chinese Society of Clinical Oncology (CSCO) criteria and modified NIH criteria. For the proband, endoscopic full-thickness resection (EFTR) was conducted for his partial gastric GISTs in 2016, and radical gastrectomy was conducted in 2022. Another patient of this family (the proband’s niece) was treated by endoscopic submucosal dissection (ESD) in 2022. Other patients are still under follow-up.

### 5.2. Molecular Analysis

Whole-exome sequencing (WES) was conducted for finding potential genetic drivers. Genomic DNA was extracted from blood and sequenced according to standard protocols for next-generation sequencing (Novogene Co., Ltd., Beijing, China). Sanger sequencing was used to verify the suspected somatic variants identified by WES. Genomic DNA was extracted from FFPE samples of GISTs resected by EFTR and normal oral epithelial cells. PCR amplification was conducted, and the PCR products were sent for automatic DNA sequencing (Takara).

### 5.3. Cell Culture, Mutant Construction, Plasmid Transfection, and Cell Viability Assay

The HEK 293T cell line was cultured in Dulbecco’s Modified Eagle Medium containing 10% fetal bovine serum (Gibco, NY, USA) and 2 mM glutamine under a humidified atmosphere in 5% CO_2_ at 37 °C. The cell line was confirmed to be Mycoplasma-free (cat# CUL001B). KIT mutants were generated by QuikChange II Site-Directed Mutagenesis Kit purchased from Agilent Inc. (cat# 210518, Santa Clara, CA, USA). Wild-type and mutant KIT were cloned into the pcDNA3.1(+) vector using NheI and EcoRI restriction sites, respectively. The plasmids were transfected using Lipo3000 assay (ThermoFisher, New York, NY, USA, cat# L3000015). For cell-viability assays, 3000 cells were plated in 96-well plates and cultured overnight. Serum starvation was conducted for another 24 h. Compounds (imatinib, ripretinib, and avapritinib) or stem cell factor (100 ng/mL; Selleck, UT, USA) were then added in serial dilutions. Cellular ATP levels were determined after 48 h by the Cell Titer-Glo^®^ Luminescent Cell Viability Assay (cat# G7570, Promega, Madison, WI, USA). The absorbance of the plates was measured on a THERMO max microplate reader.

### 5.4. Western Blotting

Immunoblotting was carried out using standard techniques. In brief, the cells were lysed in ice-cold 1X RIPA lysis buffer, and protein concentrations were determined. Aliquots (50 μg) of protein were denatured in Laemmli loading buffer and separated on precast 4–10% NuPAGE Novex 4–12% Bis-Tris Protein Gels (NP0323, Life Technologies, Carlsbad, CA, USA). Proteins were transferred to polyvinylidene difluoride membranes, which were blocked and probed with primary antibodies and then detected using appropriate horseradish peroxidase-labeled secondary antibodies. Primary antibodies used in the study were as follows: AKT (cat#9272, CST), Phospho-Akt (Thr308) (cat#4056, CST), Phospho-Akt (Ser473) (cat#4060, CST), p70 S6K (cat#9202, CST), Phospho-p70 S6K (Thr421/Ser424) (cat#9204, CST), Stat3 (cat#4904, CST), Phospho-Stat3 (cat#9145, CST), c-Kit (cat#3074, CST), Phospho-c-Kit (Tyr703) (cat#3073, CST), Phospho-c-Kit (Tyr719) (cat#3391, CST), Erk1/2 (cat#4695, CST), and Phospho-Erk (Thr202/Tyr204) (cat#4370, CST). Proteins were visualized using enhanced chemiluminescence (Pierce, Thermo-Fisher) on Hyperfilm (GE Healthcare, New York, NY, USA).

### 5.5. Statistical Analysis

Statistical analyses were carried out using Microsoft Excel software and GraphPad Prism (GraphPad Inc., La Jolla, CA, USA). Student’s t-test and one-way analysis of variance were used to analyze the differences between two groups and among multiple groups, respectively. Experiments were repeated in triplicate. A *p*-value < 0.05 was considered statistically significant in all cases and is indicated by one asterisk. A *p*-value < 0.01 and <0.001 is represented by two and three asterisks, respectively. Error bars shown in the figures represent standard deviations.

## Figures and Tables

**Figure 1 ijms-24-00830-f001:**
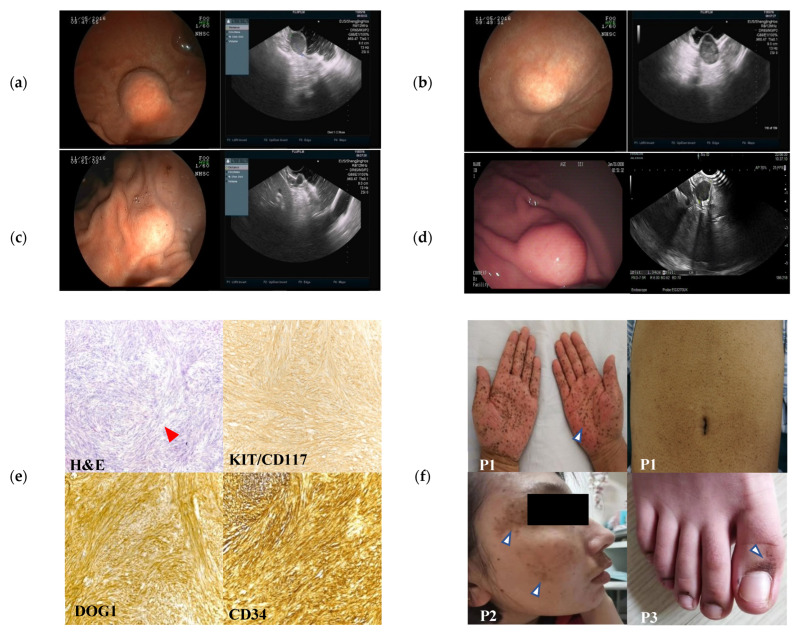
Clinicopathological presentations and molecular analysis of familial GISTs with hyperpigmentation. (**a**) Endoscopic and endoscopic ultrasound (EUS) images of the proband showing the lesion at the gastric greater curvature (23 × 20 mm) originating from the muscularis propria. (**b**) Endoscopic and EUS images showing a lesion located at the lesser curvature (30 × 20 mm) originating from the muscularis propria with echogenic heterogeneity, regular margins, and exophytic growth. (**c**) Endoscopic and EUS images of the proband showing a lesion at the lesser curves (10 × 8 mm) originating from the muscularis propria with echogenic heterogeneity and regular margins. (**d**) Endoscopic and EUS images of patient 2 (P2; niece of the proband) showing a lesion in the fundus (12 × 9 mm) originating from the muscularis propria with homogeneous echogenicity and regular margins. (**e**) Hematoxylin and eosin staining of the proband’s largest lesion, revealing bundles of spindle cells arranged in fascicles (red arrow). Tumor cells display positive immunoreactivity for KIT/CD117, DOG1, and CD34 (original magnification, ×20). (**f**) Cutaneous hyperpigmentation of three family members (P1, P2, and P3). (**g**) Sanger sequencing of the proband’s tumor DNA, revealing a G-to-C transversion at codon 829 of *KIT* exon 18, which resulted in an A829P amino acid change. The same *KIT* mutation was identified in oral epithelial cell DNA (normal), indicating a germline origin. (**h**) Targeted Sanger sequencing of three other family members with hyperpigmentation revealed the same mutation.

**Figure 2 ijms-24-00830-f002:**
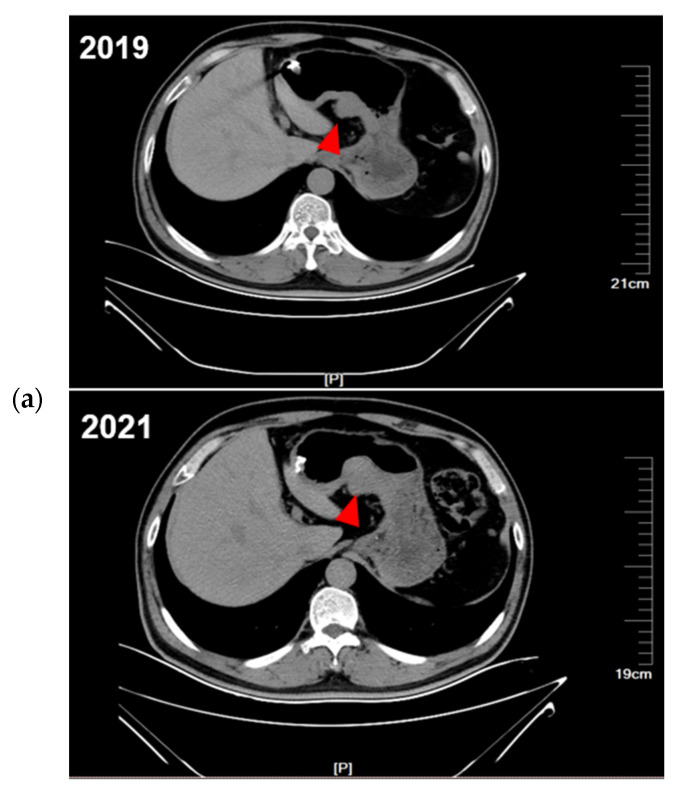
The diagnostic and therapeutic process of this family. (**a**) CT showed continued progression of the proband’s GISTs (red arrows). (**b**) The overall timeline of the entire follow-up process.

**Figure 3 ijms-24-00830-f003:**
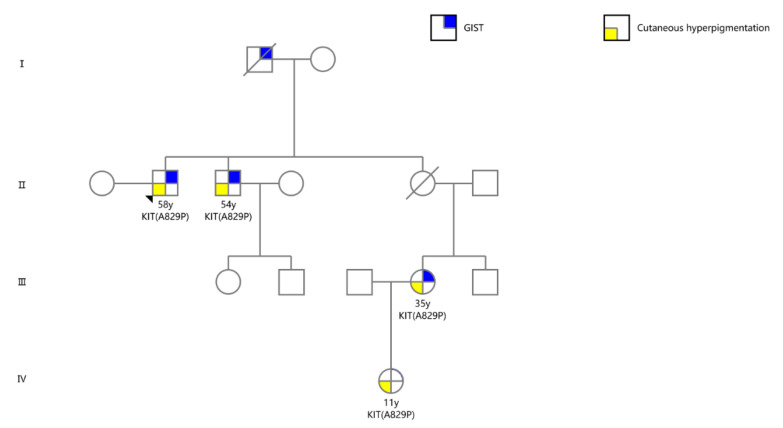
The family pedigree.

**Figure 4 ijms-24-00830-f004:**
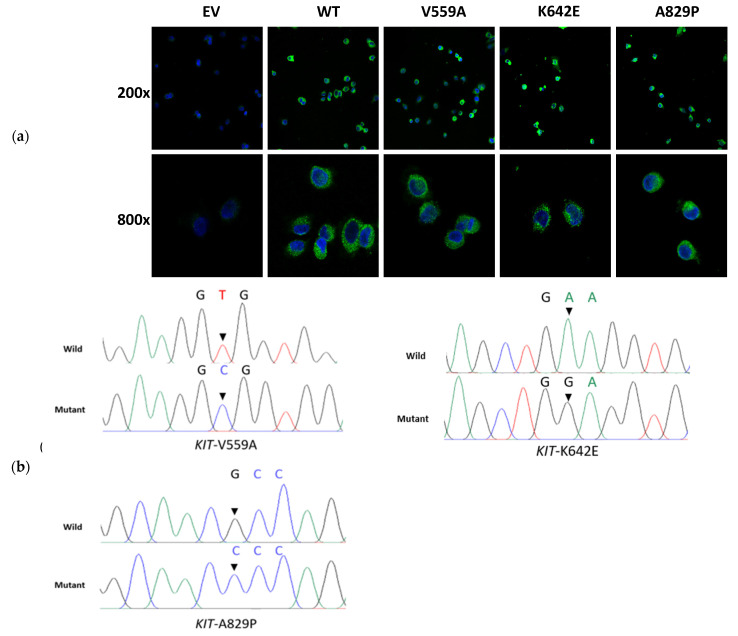
The successful construction of wild/mutant *KIT* overexpressed cells. (**a**) Immunocytochemistry images showing the expression of KIT (green; nuclei: blue) in 293T cells. (**b**) Sanger sequencing identified that mutant *KIT* had been successfully constructed and transfected into 293T cells. The total RNA was extracted from transfected cells, and reverse transcriptions were subsequently conducted. PCR experiments were used for amplifying targeted sequences. (**c**) Western blotting identified that the wild/mutant KIT proteins were successfully expressed in stably transfected 293T cells.

**Figure 5 ijms-24-00830-f005:**
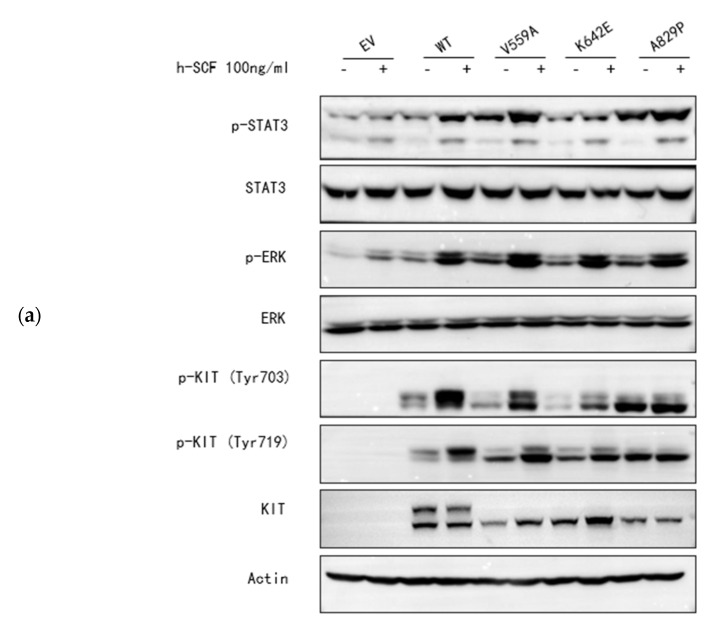
Ligand-independent activations and target drug sensitivities of mutant KIT proteins. (**a**) Mutant KIT proteins showed ligand-independent phosphorylation in the absence of stem cell factor (SCF, 100 ng/mL) stimulation. 293T−KIT/WT, 293T−KIT/V559A, 293T−KIT/K642E, and 293T-KIT/A829P were serum-starved for 24 h and treated with or without 100 ng/mL SCF for 5 min. Total cell lysates were separated by SDS–PAGE, electrotransferred to Immobilon P membrane, and probed with antibodies. (**b**) KIT−A829P significantly promoted cell proliferation and imatinib resistance in the CCK−8 assay. Briefly, 3000 cells were plated in 96-well plates and cultured overnight. Serum starvation was conducted for another 24 h, and drugs (imatinib−250 nM, ripretinib−150 nM, avapritinib−40 nM) were then added in serial dilutions. CCK8 assays were conducted for another 48 h; * *p* < 0.5, ** *p* < 0.01, *** *p* < 0.001, **** *p* < 0.0001. (**c**) Ripretinib significantly inhibits the auto-activation of aberrant KIT protein. 293T−KIT/WT, 293T−KIT/V559A, 293T−KIT/K642E, and 293T−KIT/A829P were serum-starved for 24 h and treated with or without targeted drugs (imatinib−250 nM, ripretinib−150 nM, and avapritinib−40 nM) for another 48 h. Western blotting was used for detecting the phosphorylation conditions of associated proteins.

## Data Availability

To protect the privacy of the patients, all data related to the patients cannot be made available for public access, but all sequencing results from this manuscript are secured at Shengjing Hospital and are available from the corresponding author (guojt@sj-hospital.org) upon reasonable request and approval by the Ethics Review Committee.

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
