# Peer review of "Chinese Pedigree with Hereditary Gastrointestinal Stromal Tumors: A Case Report and Literature Review"

_ijms, 2023, doi:10.3390/ijms24010830_

Round 1

Reviewer 1 Report

The article “Chinese pedigree with hereditary gastrointestinal stromal tumors: a case report and literature review” reports a case of familial gastrointestinal stromal tumor (GIST) accompanied with cutaneous hyperpigmentation and the rare A829P mutation of KIT. The authors suppose that this is the first case of hereditary GIST originating from a germline A829P KIT mutation. Via cell models overexpressing various primary driver mutations of KIT, the authors performed a quick screening of three candidate drugs and the activity of downstream molecular pathways including JAK/STAT, MAPK/ERK and PI3K/AKT.

The case report is well-written and comprehensible. However, the format is often unsatisfactory (e.g. heading in the abstract; position of figure headings). The major issue is the lack of the entire materials and methods. It is mandatory to add a materials methods section.

Specific comments:

1.    The entire materials and methods section is missing. Please add it to the revised version of the manuscript.

2.    The figure captions are insufficient. Please describe in the captions what was done in the corresponding experiment. The interpretation of the experiments should be done in the main text. Please move the headings of each figure below the respective figure right before the text of the figure caption.

3.    Page 2, line 24, 28, 44: Please remove the headings in the abstract.

4.    Page 5, line 100: “… and negativity for calponin, smooth muscle actin, and S-100.” The corresponding data are missing. Please show calponin, SMA and S-100 immunohistochemical images.

5.    Why did the authors not use one or several GIST cell lines as cell model, but an embryonic kidney cell line HEK 293T for their in vitro studies? Please debate this in the text.

6.    Page 10. Figure 4A: The authors show that the transfection of KIT mutant plasmids worked as shown by fluorescence microscopy in Figure 4A. Did the authors have any information about the overexpression efficiency of KIT mutations by Sanger sequencing? This is the baseline for the subsequent cytotoxicity testing. Please debate this in the text.  

7.    Page 10, Figure 4B: Which cell viability assay was performed here? Which drug concentrations were used? Did the authors perform a dose-finding experiment prior to cell treatment?

8.    Page 10, Figure 4B: Statistical comparison of the distinct drug treatments is not shown in the graph. Please add the corresponding statistics. Please remove all “NS” from the graph for simplification reasons.

9.    Have the authors tried any combination treatments with imatinib, ripretinib and avapritinib? It would be interesting to see potential synergistic effects on PI3K/AKT, JAK/STAT and MAPK/ERK pathway activation in the different KIT mutants.

10.  Page 11, Figure 4C: What is the rational of the h-SCF treatment? Please debate in the text.

11.  Page 11, Figure 4D: Which drug concentrations were used? How many hours were the cells treated with the drug? Please add a description of the method in a materials and methods paragraph.

Reviewer 2 Report

Familial gastrointestinal stromal tumor (GIST) is a rare autosomal dominant genetic disorder that has only affected a few families so far. This study discovered a rare KIT mutation in familial GISTs that caused cell progression and TKI resistance due to constitutive KIT activation. Likewise, drug-sensitivity studies revealed that ripretinib administration may be the most appropriate targeted therapy for this family.

Overall, this study is well-written and well-organized, the methodology appears to be correct and the references are up to date. The topic is very interesting, and the results provide the readers with very useful information for better therapeutic management of these difficult tumors.

Congrats to the authors on this intriguing case.

Reviewer 3 Report

This study describes the familial GISTs with hyperpigmentation in the skin showing newly identified KIT mutation (c.G2485C) both in the gastric tumor and normal oral tissues. However, the skin-pigmented lesions were not examined by sequencing. The article is interesting but seems to be hesitant to submit from other journal styles without correction.

The flows of context are redundant and somewhat insistent.

1.      (abstract line 29): the patient is male or female?

2.      (abstract, line 33-34): Does cutaneous pigmentation with GISTs suggest the possibility of familial hereditary disease? The sentence seems to be too strong to confirm in the abstract. The authors did not examine the sequencing or NGS in the pigmented skin lesion, though.

3.      (Introduction line 62-63) Authors described the immunohistochemical results of Kit, DOG-1, and CD34 in the case presentation and Figure 1. You need to explain the diagnostic immunohistochemical criteria in these lines. How much Kit, DOG-1, CD34 rate is positive? Please add the reference.

4.      Line 69: Please add the reference.

5.      Line 85: how many nodules did the patient have exactly? You continued to vaguely describe “multiple”.

6.      Line 103: what risk stratification of GISTs did the patient have? The tumors seem to be low grade, and then why did the clinicians offer the imatinib?? Need to provide a guideline for the treatment.

7.      Line 113: Because the patient had enlarged gastric subepithelial lesions, did he undergo total gastrectomy? Is there any guideline for surgery for total gastrectomy in cases of GISTs without a high grade? What size of gastric GIST is big to get surgery of total gastrectomy? I cannot access the supplementary Figure.

8.      Figure 1(h): it is not offensive, but this picture is suspicious for all the same sequencing profiles. They have all same lines, heights, and angles in all P2, P3, and P4. Could you please provide the original sequencing data in the reply sheet?

9.      Please add the reference at the end of the sentence throughout the manuscript.

10.  Figure 1: (e) pictures, immunohistochemical names are in mistake. Please revise.

Round 2

Reviewer 1 Report

Review of the revised article IJMS_2090840

The authors have satisfactorily addressed the reviewer’s comments. The manuscript has been improved and is worth being published after providing the mentioned PDF files for supplementary figures S1 and S2 and after implementation of the following minor comments:

Minor issues:

1.     Page 6, line 122; page 7, line 146: Supplementary Figure S1 (IHC images) and S2 (image of gross tumor) were mentioned in the text. However, the corresponding PDF files were not attached to the revised manuscript. Please add them to the final revised version.

2.    The appendix file “original images” contains original Western Blot images presented in the revised Figure 4C, Figure 5A and Figure 5C. However, the Figure numbering in the appendix file still matches with the original manuscript. Please correct the following figure headers:

-       Appendix file page 1: Figure 5C

-       Appendix file page 2: Figure 5A

-       Appendix file page 3: Figure 4C

Moreover, please rearrange the figures in the appendix file in a chronological order.

3.    Page 14, line 276: “(a) Immunocytochemistry images showing the expression of KIT in 293T cells;…” Please change this to “(a) Immunocytochemistry images showing the expression of KIT (green; nuclei: blue) in 293T cells;…”

4.    Page 15, line 327: The bar graph includes also significances indicated by * or ****. Please add the corresponding P values *P < 0.5 and ****P < 0.0001.

Author Response

Thanks for your suggestions for the paper. We have revised the manuscript according to your requirements. Besides, the supplementary figures and the appendix file “original images” will be resubmitted.

Reviewer 3 Report

Well done!

Author Response

Thanks for your efforts and suggestions for the paper! We have resubmitted the revised manuscript.